# Polymeric Nanoparticles with Embedded Eu(III) Complexes as Molecular Probes for Temperature Sensing

**DOI:** 10.3390/molecules27248813

**Published:** 2022-12-12

**Authors:** Kirill M. Kuznetsov, Vadim A. Baigildin, Anastasia I. Solomatina, Ekaterina E. Galenko, Alexander F. Khlebnikov, Victor V. Sokolov, Sergey P. Tunik, Julia R. Shakirova

**Affiliations:** 1Department of General and Inorganic Chemistry, Institute of Chemistry, St. Petersburg State University, St. Petersburg 198504, Russia; 2Department of Organic Chemistry, Institute of Chemistry, St. Petersburg State University, St. Petersburg 198504, Russia

**Keywords:** luminescent europium complexes, lifetime temperature sensitivity, latex nanospecies, phosphorescence lifetime imaging, intracellular localization, cell thermometry

## Abstract

Three novel luminescent Eu(III) complexes, **Eu1**–**Eu3**, have been synthesized and characterized with CHN analysis, mass-spectrometry and ^1^H NMR spectroscopy. The complexes display strong emission in dichloromethane solution upon excitation at 405 and 800 nm with a quantum yield from 18.3 to 31.6%, excited-state lifetimes in the range of 243–1016 ms at 20 °C, and lifetime temperature sensitivity of 0.9%/K (**Eu1**), 1.9%/K (**Eu2**), and 1.7%/K (**Eu3**). The chromophores were embedded into biocompatible latex nanoparticles (**NPs_Eu1**–**NPs_Eu3**) that prevented emission quenching and kept the photophysical characteristics of emitters unchanged with the highest temperature sensitivity of 1.3%/K (**NPs_Eu2**). For this probe cytotoxicity, internalization dynamics and localization in CHO-K1 cells were studied together with lifetime vs. temperature calibration in aqueous solution, phosphate buffer, and in a mixture of growth media and fetal bovine serum. The obtained data were then averaged to give the calibration curve, which was further used for temperature estimation in biological samples. The probe was stable in physiological media and displayed good reproducibility in cycling experiments between 20 and 40 °C. PLIM experiments with thermostated CHO-K1 cells incubated with **NPs_Eu2** indicated that the probe could be used for temperature estimation in cells including the assessment of temperature variations upon chemical shock (sample treatment with mitochondrial uncoupling reagent).

## 1. Introduction

The growing interest in noninvasive thermometry with high spatial and temporal resolution is largely due to the importance of these kind of measurements for understanding critical features of physiological processes at the cellular level [1,2,3]. Among various techniques used in this area, luminescent microscopy based on different thermosensitive probes is one of the most reliable and convenient methods because of the fast and reversible response to temperature changes and high spatial resolution up to a few hundred nanometers [4,5,6]. Small-molecule luminescent probes suitable for nanothermometry can be divided into three major classes [7] depending on the key characteristics of their emission: fluorescent [8,9] and phosphorescent [10,11] thermosensitive chromophores with excited-state lifetime in the nanosecond and microsecond domains, respectively, and lanthanide compounds [7,12,13,14,15], which display lifetimes in the range from several hundred µs to 2 ms. These types of probes are now widely used for temperature measurements in bio-samples of different nature (see reviews [1,5,6,16] and some recent research papers [17,18,19]. However, the application of these probes in biomedical studies faces some problems characteristic of the certain types of emitters that are a consequence of the specific features of their photophysical properties. Fluorescent compounds constitute the largest group of thermosensors potentially applicable in biological studies [8], which suffer from low bleaching stability and the biasing effect of the sample background emission interfering with the probe signal both in ratiometric and lifetime-based modalities [20]. In addition, the dynamic interval of a probe fluorescence lifetime is limited to several nanoseconds, which complicates the measurements in the lifetime domain technically and makes the corresponding equipment rather expensive [20]. The phosphorescent thermosensitive transition metal complexes of Ru(II) [21,22], Ir(III) [23], and Pt(II) [24,25] are largely free from the disadvantages of the fluorescent emitters mentioned above but they naturally display a strong effect of oxygen quenching onto their emission parameters that results in the crosstalk between two environmental characteristics (e.g., T-O_2_), which may considerably distort the results of temperature sensing. The thermosensors based on lanthanide complexes [7,20,26,27,28] display a large Stokes shift and a wide dynamic interval of lifetimes, show no emission quenching with molecular oxygen, and demonstrate high emission and lifetime response to temperature variations in physiologically relevant intervals, of which Eu(III) organometallic compounds [29] are the most promising candidates for temperature measurements. Nevertheless, the most significant disadvantage of this type of sensor for bioimaging applications is extremely strong emission quenching in aqueous media. The quenching mechanism consists of vibrational energy transfer to water molecules [30]; therefore, the isolation of these chromophores from interaction with the solvent by embedding into water-impermeable polymeric nanospecies is an approach of choice in the design of effective europium temperature sensors.

In our previous study [31], we used nanoemulsion polymerization to incorporate thermosensitive Eu(III) complexes into latex nanoparticles (NPs) that effectively prevented europium emission quenching. The obtained NPs displayed a rather high lifetime response to temperature variations (a sensitivity up to 0.84%/K and resolutions of 0.26 K) and complete reversibility in cycling experiments that made them promising for application in biomedical studies. However, the emission of these europium chromophores can be excited only with UV irradiation at 365 nm which is far beyond the biologically friendly interval, and the chemical composition of the NPs does not imply their vectorization for the targeted delivery of the sensor to a certain cell compartment or tissue. In the present study, we applied the other europium complexes, which allowed emission generation upon excitation either in the visible range of spectrum (405 nm) or with two-photon excitation (800 nm) that made the nanothermometers applicable in prolonged biological experiments without damage to the samples under study. The photophysical properties of the obtained NPs were studied in detail along with investigation of their lifetime sensitivity to temperature in different media. We also modified the polymerization technique by using 2-aminoethyl methacrylate as a component of the polymerization reaction that results in the NPs’ surface amination and opens the way to the NPs’ vectorization, e.g., with a phosphonium cation. The suitability of the NPs for temperature mapping in cells using phosphorescent lifetime measurements (PLIM) was tested in CHO-K1 cell culture.

## 2. Results and Discussion

### 2.1. Synthesis of Thermosensitive Eu(III) Complexes

In this study, the choice of the ligands in the Eu(III) ion environment was dictated by the necessity to shift the excitation wavelength of the final complexes into a biologically friendly interval, at least to the wavelengths >400 nm, that would make it possible to use a “blue” 405 nm laser, which is a standard component of luminescent microscopy equipment. To meet this requirement, we introduced into the coordination sphere of europium complexes the ligands (L_1_L_3_, Figure 1) containing developed aromatic systems that, as a rule, results in a bathochromic shift of the absorption/excitation wavelength. Indeed, absorption and excitation spectra of **L_1_**–**L_3_** are extended well below 400 nm (see Appendix A), thus giving a chance to generate emission of the corresponding europium complexes with the 405 nm laser. These ligands also contain spatially separated donor/acceptor diads that make them promising from the viewpoint of two-photon absorption and the possibility to be excited by using near-infrared irradiation at about 800 nm. This type of low-energy excitation is even more promising for the microscopic studies of biological samples. **L_1_** [32,33] and **L_2_** [34] were obtained according to the published procedures; **L_3_** is a novel diimine ligand based on the imidazophenanthroline aromatic system with an electron-withdrawing NO_2_ group in a phenyl substituent. The synthesis and characterization of **L_3_** is described in the experimental section; its synthetic scheme, proton NMR, and ESI^+^ mass-spectra are given in Appendix A—see Appendix A. 

**Eu1**, **Eu2**, and **Eu3** complexes were synthesized using the standard procedures for the preparation of europium tris-chelates [28] and [Eu(TTA)(diimine)] [34] emitters; see Figure 1. The target compounds were obtained in a good yield, and purified and characterized using elemental analysis, HR ESI^+^ mass-spectrometry (Appendix A), and ^1^H NMR spectroscopy (Appendix A). The positions of the signals in mass spectra and their isotopic distributions were in excellent agreement with the suggested stoichiometry of the obtained compounds. The ^1^H NMR spectra displayed a low-field shift of the corresponding proton signals typical for Eu(III) complexes; the number of signals and their relative intensity and multiplicity fit well with the structural patterns shown in Figure 1 and allowed the complete assignment of the resonances observed (Appendix A).

### 2.2. Photophysical Properties of Eu(III) Complexes

The obtained **Eu1**–**Eu3** complexes luminesce in dichloromethane solution; their absorption, excitation, and emission spectra are shown in Figure 1 and Appendix A, and the major photophysical characteristics are summarized in Table 1.

In dichloromethane (DCM) solution, all complexes displayed emission bands typical for Eu(III)-based chromophores with the major component located at 612 nm. The emission of **Eu1** and **Eu2** may be excited with a 405 nm laser that makes possible their use in imaging experiments without photodamage to the sample under study. Moreover, both complexes showed appreciable two-photon absorption under excitation at 800 nm, thus increasing their suitability for bioimaging experiments. **Eu2** also displayed independent fluorescence of coordinated **L2** (Figure 1) that, in principle, provides an opportunity for ratiometric measurements. On the contrary, **Eu3** gave appreciable emission intensity only by using the ultraviolet 365 nm laser radiation that considerably limits its application in the studies of biological objects. In DCM, the complexes displayed rather high emission quantum yields ranging from 18.3 to 31.6% and lifetimes in the interval from two hundred µs to one ms (Table 1). The lifetimes of **Eu1**–**Eu3** demonstrated a substantial temperature dependence in the physiologically relevant interval (see Figure 1b, Appendix A) which can be approximated by Appendix A; see Appendix A [4,35].

These experimental data also allow calculating the emitters’ temperature sensitivity (*S_T_*, Appendix A) and relative temperature sensitivity (*S_r_*, Appendix A); the latter showed the highest value of 1.9%/K for **Eu2** that makes the complexes promising for application in nanothermometry. To avoid europium emission quenching in the aqueous/physiological media, we used an approach successfully applied in our previous study [31], namely, the incorporation of these emitters into latex nanoparticles.

### 2.3. Synthesis of Latex Nanoparticles (NPs)

The first generation of thermosensitive NPs described in our previous publication [31] demonstrated that europium complexes’ incorporation into latex nanoparticles prevents emission quenching in aqueous media and also makes their emission characteristics completely reversible in thermocycling experiments at least during nine cycles. The NP positive charge brought on by the surfactant, cetyltrimethylammonium bromide (CTAB), essentially facilitates adsorption onto cell membranes and subsequent probe internalization [36]; however, the absence of surface functional groups suitable for conjugation with vectors makes the probe uptake non-targetable, giving a random probe distribution in cells that significantly limits their application in bioimaging. Therefore, we prepared the second generation of luminescent NPs (**NPs_Eu1**–**NPs_Eu3**) through the introduction of primary amino groups onto the NPs’ surface via the use of 2-aminoethyl methacrylate (AEMA) in the polymerization protocol (Figure 2).

The numerical photophysical and physicochemical data for **NPs_Eu1**–**NPs_Eu3** are given in Table 1 and Table 2, respectively; the absorption, excitation, and emission spectra are shown in Appendix A. The excitation and emission band profiles of the europium-containing nanospecies in aqueous dispersions were essentially similar to those obtained for the parent europium complexes in a DCM solution. However, the emission quantum yields for all NPs displayed a ca. 40% reduction compared to the corresponding complexes in DCM, that may be due to partial emission quenching by solvent molecules or is related to excitation energy transfer to the polymeric matrix. Nevertheless, the emission intensity was still rather high, particularly taking into account the aqueous environment of the chromophores. It is also worth noting that the lifetime sensitivity of the NP probes was reduced to a different extent depending on the nature of the europium emitter; the strongest decrease was observed in the case of **NPs_Eu3**, whereas **NPs_Eu2** displayed an appreciable *S_r_* magnitude equal to 1.3%/K, still the highest in this group of nanosensors.

The diameter of the **NPs_Eu1** was found to be 116 nm with a PDI of 0.07 (Table 2; Appendix A) as measured using DLS, whereas the average diameter determined using transmission electron microscopy (TEM) with the negative staining was 85 nm (Table 2, Figure 2a). The latex NPs containing **Eu2** and **Eu3** showed essentially similar size characteristics: the NPs’ size equaled ca. 100 nm and had a PDI of 0.07 and 0.14, respectively (Table 2; Appendix A). A similar decrease in the particle diameter was observed using TEM: the size reduced to 72 and 70 nm for **NPs_Eu2** and **NPs_Eu3**, respectively (Table 2; Figure 2b,c). The difference between the DLS and microscopic measurements is related to the formation of a hydrate shell around the particles in the aqueous dispersions and its shrinking in a dry state [37].

All NPs possessed positive ζ-potentials due to the presence of positively charged quaternary and primary amino groups from CTAB and AEMA, respectively, and imidazoline functions of the VA-044 initiator on the surface. In particular, the ζ-potential of **NPs_Eu1** equaled 40 mV, 31.6 mV for **NPs_Eu2**, and reached only 22.4 mV for **NPs_Eu3** (Table 2; Appendix A). The magnitude of the ζ-potential and wider size distribution of **NPs_Eu3** compared to **NPs_Eu1** and **NPs_Eu2** evidence a weak colloid stability of the former that can lead to the aggregation in the presence of biological molecules or upon an increase in media salinity. These observations, together with the necessity of the **Eu3** excitation with UV laser radiation and the lowest *S_r_* value, indicate that its applicability is severely limited in the studies of biological objects.

The next step in the NPs’ structure investigation was the estimation of the complexes’ location within the nanoparticles. To understand how deep the Eu emitters were localized inside of the NPs, we carried out TEM without negative staining that allows detecting the heavy elements such as europium. The size of the NPs obtained from the TEM pictures is shown in Figure 2d–f and Table 2. All NPs demonstrated a size smaller than that observed in the experiments with negative staining by uranyl acetate (UrAc): the decrease in NPs’ diameters equaled 9–10 nm. A comparison of the results obtained with and without negative staining indicates that emitter molecules were localized in the NPs under the ca. 5mm-thick surface layer. Therefore, the outer layer containing a cross-linked polymer structure blocked the water permeability into the core which prevented luminescence quenching for at least 1.5 years that we observed during the development of the sensor. These data are in a good agreement with those presented in our previous article, where X-ray photoelectron spectroscopy was used to demonstrate the absence of europium in the surface layer [31].

We also calculated the number of complexes per particle using the data on the NPs’ size and their composition determined by ICPOES (Table 2, Appendix A) that gave the average value from 350 to 800 molecules of Eu complexes per particle. Interestingly, the number of the complex molecules in **NPs_Eu2** was ca. two times higher than the magnitudes obtained for the other species. The reason for this difference is not clear and requires the investigation of the polymerization mechanism in more detail. However, the higher amount of **Eu2** per particle can be favorable for biological experiments as it increases the brightness of phosphorescence, decreases the amount of NPs needed for each experiment, and reduces the signal accumulation time.

To finally choose the most promising candidate for biological studies, we estimated the NPs’ lifetime sensitivity in a physiological temperature range from 20 to 40 °C. **NPs_Eu1** and **NPs_Eu2** were only used in these experiments as the application of **NPs_Eu3** in biological studies is limited due to the necessity of excitation with UV laser radiation and low colloid stability as mentioned above. The dependence of lifetime on temperature for **NPs_Eu1** and **NPs_Eu2** is presented in Appendix A. The lifetime of **NPs_Eu1** varied in the range 1020–870 µs, whereas the lifetime of **NPs_Eu2** changed from 520 to 430 µs at 20 and 40 °C, respectively, which points to a substantially shorter time for the data acquisition in PLIM experiments in the latter case. The whole set of data obtained for the **NPs_Eu2** particles—highest temperature sensitivity, emission excitation with visible light, high colloidal stability, and shortest lifetime among the studied NPs—indicates that the **NPs_Eu2** is the most promising probe for spatial and temporal temperature monitoring in biological systems by using the PLIM modality. These considerations prompted us to study in detail the dependence of the **NPs_Eu2** probe lifetime on the temperature between 20 and 40 °C in water, Dulbecco’s phosphate buffer (DPB), and in the model physiological media (Dulbecco’s modified eagle medium (DMEM) supplemented with 10% fetal bovine serum (FBS)), which simulates the microenvironment of the probe in biological samples to a maximum extent; see Figure 3. 

The data given in Figure 3a show that the calibration curve built up for the water solution differed substantially to those obtained in DPB and the DMEM + 10% FBS mixture, whereas the data for the two latter media are very close each other. The difference in the probe behavior observed for water and solutions with substantial salinity looks systematic and can be explained by the electrostatic interaction of the NPs with cations that may change the emitter lifetime response to variations in the media temperature. However, this difference falls in the limits of experimental uncertainty (ca. 5%) that make it possible to use all three sets of data for the calibration of lifetime vs. temperature dependence. Additionally, we checked the effect of the other environment stimuli (pH and viscosity; see Figure 3a) that showed a lifetime independence on these media characteristics. Thus, all obtained data have been used to build up a general calibration curve shown in Figure 3b which gave the relative temperature sensitivity of the sensor 1.3%/K. In the last decade, several lifetime-based temperature sensors have been obtained and applied for in vitro and in vivo biological experiments; see data in Appendix A [31,38,39,40,41,42,43,44,45,46]. Two of them displayed a strong temperature sensitivity of ca. 7.5%/K [38] and 6.3%/K [39] but they were not without rather serious drawbacks. The nanocapsule sensor, based on excitation energy transfer from a palladium phthalocyanine photosensitizer onto a perylene emitter [38] through a chemical reaction, displayed an extremely long lifetime (second range!), which was eventually determined by the reaction rate constant. These photophysical characteristics suit in vivo temperature measurements well but cannot be applied for cell experiments in PLIM mode because of the unacceptably long image acquisition. The other effective temperature-sensitive probe (HPS/Butter/DSPE-PEG-Biotin nanorods) [39] was prepared by using a relatively simple technology but needed excitation with UV irradiation and showed a lifetime in the nanosecond domain that required rather expensive equipment for lifetime measurements. The other temperature sensors working in lifetime mode [31,40,41,42,43,44,45,46] displayed a sensitivity comparable to or even lower than that obtained for **NPs_Eu2**. These observations, together with the possibility to use **NPs_Eu2** in PLIM mode to acquire the sensor signal, indicate that this probe is promising for application in cell experiments.

Cyclic temperature measurements between 20 and 40 °C (Figure 3c) were carried out in water dispersion and a DMEM + FBS solution. Similarly to the behavior of the calibration curves described above, these experiments gave different absolute lifetime values for water and DMEM + FBS solutions but displayed very good reproducibility that is of critical importance for temperature monitoring in living objects under different conditions and actions of outer stimuli. 

### 2.4. Cell Experiments

#### 2.4.1. Cytotoxicity and Localization of Probe in Cells

To further assess the applicability of **NPs-Eu2** in biological studies, we carried out cytotoxicity tests, and measured the dynamics of the probe uptake and its subcellular distribution in living CHO-K1 cells. The results of the MTT assay are shown in Figure 4a. The data indicate that the viability of cells was high, up to ca. 10^−2^ wt.% of dry residue. A further increase in the concentration led to a significant increase in the probe toxicity. Therefore, in order to minimize the toxic effect of the probe and maximize the luminescent signal, for subsequent bioimaging experiments, we chose a probe-limiting concentration of 0.0195 wt.% of dry residue or 8.56 × 10^−7^ mmol in terms of the amount of **Eu2** (calculations were performed using Appendix A; see Appendix A). 

The dynamic of internalization of **NPs_Eu2** in living CHO-K1 was monitored using confocal microscopy with excitation at 405 nm in two emission channels: 500–550 nm and 570–620 nm (Appendix A). The emission of the probe could be detected in the cell cytoplasmic region already after 1 h of incubation. The emission intensity, however, grew slowly, reaching a maximum after 24 h of incubation. Based on the results of cytotoxicity and the dynamics of probe uptake, a probe concentration of 0.0195 wt.% and incubation time of 24 h were chosen for further experiments.

To reveal the localization of the probe in cellular compartments, CHO-K1 cells were co-stained with **NPs_Eu2** and the commercial dyes LysoTracker Deep Red and BioTracker 405 Blue, which are fluorescent probes for acidic organelles and mitochondria, respectively. Unfortunately, the blue fluorescence of the ligand (see above) overlapped the mitochondria tracker signal which prevented the assessment of the probe accumulation in this cellular compartment. Nevertheless, confocal microscopy of CHO-K1 cells co-stained with **NPs_Eu2** and LysoTracker Deep Red (Figure 5) showed that the probe was predominantly localized in acidified cell compartments (endosomes of different types and lysosomes) that most probably indicates the probe uptake by endocytic mechanisms or micropinocytosis [47].

#### 2.4.2. PLIM Experiments

As an evaluation of applicability of the nanoparticles obtained for intracellular temperature measurements, a series of PLIM experiments were carried out. In the first one, living CHO-K1 cells stained with **NPs_Eu2** were thermostated for 40 min prior to microscopy and during the experiment at 30 °C and 40 °C; Figure 6.

The elevation of temperature by 10 degrees (from 30 to 40 °C) led to a decrease in the probe lifetime by ca. 50 µs (from 480 to 430 µs). The maxima of lifetime distribution fit well the calibration curve data obtained in the solution (Figure 3b). It is important to note that the lifetime distribution across the whole image is rather broad with a width at half-height of ca. 25 µs that can be assigned to the temperature variation in different compartments of the living cells as well as to the accuracy of lifetime measurements (5%, see above). Thus, the obtained results indicate that the probe can be used for measurements of temperature in the cells.

In the second series of PLIM experiments, we monitored the probe lifetime variations as a measure of cell physiological response to the action of chemical *stimuli*, namely, the addition of mitochondrial uncoupling agent (carbonyl cyanide-4-(trifluoromethoxy) phenylhydrazone, FCCP), which usually gives an increase in cell temperature [48,49,50,51] due to the inhibition of ATP synthesis in the mitochondria and the induction of energy release in the form of heat [46,52]. 

Analysis of the lifetime distribution across the studied images, see Figure 7, indicates that the probe lifetime gradually dropped down after the addition of FCCP for ca. 20 µs, indicating an increase in intracellular temperature by ca. 3–4 °C, which is close to the values found in the similar cell experiments [51].

### 2.5. Modification of Latex NPs’ Surface

As mentioned above, modification of the NPs’ surface with primary amino groups makes possible the sensor vectorization using a standard chemistry of amide bond formation for the attachment of vector functions to the sensor. To demonstrate the applicability of this approach, we modified the **NPs_Eu2** surface with a phosphonium cation using N-hydroxysuccinimidyl ester of 2-carboxyethyl triphenylphosphonium bromide (**TPP-NHS**); see Figure 2. The phosphonium cation is a well-known vector for the delivery of molecular objects to the mitochondria, which are the “powerhouse of the cell” [53,54] and may show considerable variations in local temperature [55]. The details of the synthesis and characterization of modified species (**NPs_Eu2_TPP**) are given in the Materials and Methods section and Appendix A. The successful conjugation of **TPP** was confirmed by the analysis of phosphorus content in the NPs using ICPOES, which showed the presence of **TPP** in the system in the amount of 22,800 molecules per **NPs_Eu2_TPP** particle (Table 2). The obtained NPs displayed a size of 99 nm (measured using DLS, Table 2, Appendix A) which is nearly identical to the magnitude obtained for the **NPs_Eu2** species, but the ζ-potential was decreased to 24 mV (Appendix A) compared to the initial nonmodified species due to a reduction in the number of surface amino groups. It was also found that the photophysical characteristics of the species modified with phosphonium cations were very similar to those of the starting nonmodified **NPs_Eu2** including lifetime sensitivity to temperature variations; see Appendix A. However, the studies on CHO-K1 cells’ incubation with **NPs_Eu2_TPP** showed that this species does not display preferential localization in mitochondria. The experiments on the probe colocalization with LysoTracker Deep Red gave the values of Pearson and Manders coefficients (Appendix A), which are even higher than those obtained for **NPs_Eu2**, which points to the essentially similar behavior of modified and nonmodified probes with respect to mitochondrial accumulation. These observations indicate that, due to so-far unknown reasons, this type of vectorization with phosphonium cations does not give the desired result and it is necessary to search for the other ways of mito-vectors’ association with the probes obtained. 

## 3. Materials and Methods

**Materials.** Methyl methacrylate (98%, MMA) and butyl methacrylate (BMA) were purchased from Acros Organics and distilled under vacuum before use. Ethyleneglycol dimethacrylate (98%, EGDMA, Sigma), 2-aminoethyl methacryate hydrochloride (AEMA, Sigma), 2,2′-azobis [2-(2-imidazolin-2-yl)-propane]dihydrochloride (97%, VA-044, Sigma), cetyltrimethylammonium bromide (CTAB), and acetone were used as received. Water was purified using a “Simplicity” (“Merck Millipore”) water purification system (type 1 water). 1,10-Phenanthroline-5,6-dione [56], Eu(OAc)_3_·nH_2_O [57], Eu(TTA)_3_·3H_2_O [58], **L1** (4-((4-(dimethylamino)phenyl)ethynyl)pyridine-2,6-dicarboxylic acid [32,33], **L2** (4-(4,6-di(1H-pyrazol-1-yl)-1,3,5-triazin-2-yl)-N,N-diethylaniline) [34], and (2-Carboxyethyl)triphenylphosphonium bromide (**CE-TPP**) [59] were synthesized according to the published procedures. Ligand **L3** has not been previously described and its synthesis is presented in this paper. Reagents (Merck KgaA, Darmstadt, Germany), general solvents (Vekton, St. Petersburg, Russia), and deuterated solvents (Carl Roth GmbH + Co. KG, Germany) were used as received without further purification.

**General Experimental Details.** Mass spectra were recorded using a Bruker maXis HRMS-ESI-QTOF, ESI^+^ or ESI- mode. ^1^H and ^1^H−^1^H COSY (400 MHz) NMR spectra were recorded on a Bruker 400 MHz Avance. Chemical shift values are reported relative to TMS (δ = 0.00). ^1^H NMR spectra were referenced to the residual signal of CDCl_3_ (7.26 ppm), Acetone-d_6_ (2.05 ppm), or DMSO-d_6_ (2.50 ppm). Microanalyses were carried out by using Euro EA3028-HT.

**Synthesis of L3,** 2-(4-nitrophenyl)-1-phenyl-1H-imidazo [4,5-f][1,10]phenanthroline.

According to the previously published procedure (https://doi.org/10.1002/ejic.202100189), 4-nitrobenzaldehyde (181 mg, 1.2 mmol, 1 eq.), 1,10-phenanthroline-5,6-dione (252 mg, 1.2 mmol, 1 eq.), aniline (223 mg, 2.4 mmol, 2.5 eq.), and ammonium acetate (185 mg, 2.4 mmol, 2.5 eq.) were added to 7 mL of acetic acid and refluxed overnight. The precipitated powder was centrifuged and washed with diethyl ether to give a yellow powder, which was then recrystallized through gas-phase diffusion of diethyl ether into dichloromethane solution of the product; yield 62%. ^1^H NMR (400 MHz, CDCl_3_, 297 K) δ = 9.24 (dd, ^3^*J_H-H_* = 4.4, ^4^*J_H-H_* = 1.8 Hz, 1H, phen), 9.14 (d, ^3^*J_H-H_* = 7.9, ^4^*J_H-H_* = 1.8 Hz, 1H, phen), 9.10 (dd, ^3^*J_H-H_* = 4.4, ^4^*J_H-H_* = 1.7 Hz, 1H, phen), 8.19 (d, ^3^*J_H-H_* = 8.9 Hz, 2H, phenylNO_2_), 7.84–7.71 (m, 6H, phenylNO_2_, phenyl, phen), 7.60 (d, ^3^*J_H-H_* = 8.9 Hz, 2H, phenyl), 7.47 (dd, ^3^*J_H-H_* = 8.4, ^4^*J_H-H_* = 1.7 Hz, 1H, phen), 7.34 (dd, ^3^*J_H-H_* = 8.4, ^4^*J_H-H_* = 4.3 Hz, 1H, phen) ppm. ES MS (*m*/*z*): [L + H]^+^ 417.1213 (calc. 418.1304). Anal. calc. for C_25_H_15_N_5_O_2_·CHCl_3_ (%): C, 58.18; H, 3.00; N, 13.05. Found: C, 58.58; H, 3.03; N, 13.29.

**Synthesis of Eu1 Complex.** In a borosilicate glass round-bottom culture tube with a screw cap (Pyrex™, 1636/24MP), ligand **L1** (100 mg, 349 mmol, 3 eq.), tetrabutylammonium hydroxide (271 mg, 698 mmol, 6 eq.), and Eu(OAc)_3_·3H_2_O (42 mg, 174 mmol, 1 eq.) were dissolved in acetone (16 mL). The solution was stirred at 50 °C overnight. The resulting mixture was extracted with dichloromethane (3 × 50 mL). Combined organic phases were dried over calcium(II) chloride and evaporated under a vacuum to give a yellow oily product; yield 70%. ^1^H NMR (400 MHz, CDCl_3_, 297 K) δ = 7.28 (s, 1H), 6.91 (d, ^3^*J_H-H_* = 8.5 Hz, 1H, phenyl), 6.46 (d, ^3^*J_H-H_* = 8.7 Hz, 1H, phenyl), 4.99 (s, 1H), 3.76–3.66 (m, 24H, NBu_4_), 2.91 (s, 6H, NMe_2_), 1.96–1.83 (m, 24H, NBu_4_), 1.56 (q, ^3^*J_H-H_* = 7.3 Hz, 24H, NBu_4_), 1.05 (m, 36H, NBu_4_) ppm. ES MS (*m*/*z*): [M-CO_2_NCO_2_]^−^ 769.0831 (calc. 769.0811). CHN analysis was obtained for potassium salt, which gives crystallizable powder suitable for analysis. Anal. calc. for C_51_H_36_EuK_3_N_6_O_12_·(H_2_O)_6_ (%): C, 47.04; H, 3.72; N, 6.45. Found: C, 47.37; H, 4.02; N 6.18.

**Synthesis of Eu2 Complex.** The complex was synthesized according to a modified literature methodology [34]. In a round-bottom culture tube, Eu(TTA)_3_·3H_2_O (119 mg, 140 mmol, 1 eq.), **L2** (50 mg, 140 mmol, 1 eq.) were dissolved in acetone (12 mL), the reaction mixture was stirred overnight at 50 °C. The precipitated yellow powder was centrifuged and washed three times with diethyl ether to eliminate the soluble ligand; yield 68%. ^1^H NMR (400 MHz, DMSO, 297 K) δ = 18.70 (s, 2H, pyr), 10.54 (s, 2H, pyr), 10.02 (d, ^3^*J_H-H_* = 8.5 Hz, 2H, phenyl), 9.40 (s, 2H, pyr), 7.51 (d, ^3^*J_H-H_* = 8.7 Hz, 2H, phenyl), 6.89 (s, 3H, thioph), 6.11 (s, 3H, thioph), 5.47 (s, 3H, thioph), 3.98 (q, ^3^*J_H-H_* = 7.2 Hz, 3H, NEt_2_), 2.92 (s, 3H, CH), 1.55 (m, 6H, NEt_2_) ppm. ES MS (*m*/*z*): [M + Na]^+^ 1199.0510 (calc. 1199.0572). Anal. calc. for C_43_H_32_EuF_9_N_8_O_6_S_3_·CHCl_3_·C_3_H_6_O (%): C, 43.11; H, 2.80; N 9.07; S 7.79. Found: C, 43.10; H, 3.04; N 8.41; S 7.24.

**Synthesis of Eu3 Complex.** In a round-bottom culture tube, Eu(TTA)_3_·3H_2_O (100 mg, 117 mmol, 1 eq.) and **L3** (50 mg, 117 mmol, 1 eq.) were dissolved into acetone (4 mL); the mixture was stirred overnight. The precipitated powder was centrifuged and washed three times with diethyl ether to eliminate the ligand, which was then recrystallized using gas-phase diffusion of diethyl ether into dichloromethane solution of the product. Light-yellow powder, quantitative yield: 57%. ^1^H NMR (400 MHz, acetone-d_6_, 297 K) δ = 14.37 (m, 1H, phen), 12.90 (m, 1H, phen), 9.36 (d, ^3^*J_H-H_* = 8.2 Hz, 1H, phen), 8.92 (d, ^3^*J_H-H_* = 8.0 Hz, 1H, phen), 8.85 (m, ^3^*J_H-H_* = 8.5 Hz, 3H, phen, phenyl), 8.67 (d, ^3^*J_H-H_* = 8.5 Hz, 2H, phenylNO_2_), 8.62 (d, ^3^*J_H-H_* = 8.5 Hz, 2H, phenylNO_2_), 8.36 (m, 3H, phenyl), 7.27 (m, 3H, thioph), 6.45 (m, 3H, thioph), 6.00 (m, 3H, thioph), 3.26 (m, 3H, CH) ppm. ES MS (*m*/*z*): [M + Na]^+^ 1256.0008 (calc. 1255.9989). Anal. calc. for C_49_H_27_EuF_9_N_5_O_8_S_3_·CH_2_Cl_2_ (%): C, 46.01; H, 2.11; N, 5.42. Found: C, 46.03; H, 2.53; N, 5.05.

**Synthesis of TPP-NHS ester.** TPP-NHS ester was synthesized according to a slightly modified literature procedure [59]. N,N’-dicyclohexylcarbodiimide (357 mg, 1.73 mol, 2.4 eq.) and N-hydroxysuccinimide (100 mg, 0.86 mol, 1.2 eq.) were added to a solution of (2-Carboxyethyl)triphenylphosphonium bromide (**CE-TPP**) (300 mg, 0.72 mol, 1 eq.), in dry acetonitrile (10 mL), and the reaction mixture was stirred at room temperature for 12 h. The reaction mixture was cooled in the fridge and filtered to remove insoluble byproducts. The solvent was removed under reduced pressure to give the product as a light-beige solid, that was used without further purification. ^31^P NMR (162 MHz, DMSO-d_6_) δ 24.40. ^1^H NMR (400 MHz, DMSO) δ 7.98–7.89 (m, 3H), 7.89–7.74 (m, 12H), 4.09–3.92 (m, 2H), 3.15–3.01 (m, 2H).

**Preparation of latex nanoparticles with embedded europium complexes.** Latex nanoparticles (NPs) were synthesized according to [31]; however, the major feature of the obtained NPs was the surface primary amino groups originating from AEMA. A typical protocol for the preparation at 5 wt.% of solids is given: 5.8 mmol of MMA, 9.4 mmol of BMA, and 2.1 mmol of EGDMA were mixed, and a europium complex (8 mg) was then dissolved in monomers using an ultrasound bath for 5 min. CTAB (290 mg), AEMA (40 mg), and V-044 (30 mg) were added to a water/acetone mixture (95:5% *v*/*v*, respectively). The water solution was heated up to 70 °C and stirred in a 100 mL round-bottom flask for 3 min. The monomer solution was then added to a reaction system and polymerization was carried out for 25 min. The crude polymer dispersion was purified by exhaustive dialysis (Orange Scientific; molecular weight cutoff = 12–14 kDa) for 3 days to remove the residual monomers and the surfactant.

**Ninhydrin assay.** The concentration of primary amino groups was estimated by the colorimetric reaction with ninhydrin [60]. A NP suspension (1 mL) was added to 500 µL ninhydrin reagent (1%, *w*/*v*) and heated in a boiling water bath for 15 min. The total volume was increased up to 5 mL and cooled to room temperature. UV absorbance was measured at a wavelength of 564 nm. AEMA was used to prepare the calibration curve.

**Conjugation of latex NPs and triphenylphosphine.** Conjugation was performed via surface amino groups of NPs and N-hydroxysuccinimidyl ester function of **TPP-NHS** as follows. 1 mL NPs (5.1 wt.%) was diluted in phosphate-buffered saline (PBS) with pH 7.4. TPP (6.68 mg, 0.024 mmol) was diluted in methanol and dropped in the PBS solution. The mixture was kept overnight at room temperature, and then, it was purified by exhaustive dialysis for 5 days to remove the residual molecules. 

**Particle size and ζ-Potential Measurements.** The size, shape, and polydispersity index (PDI) of NPs were measured using dynamic light scattering (DLS; Malvern Nano ZS, Malvern, UK) and transmission electron microscopy (TEM; Jeol JEM-2100, Tokyo, Japan). TEM pictures were obtained in two cases: with and without negative staining by uranyl acetate (UrAc). The ζ-potential in the 10^−3^ M NaCl solution was also determined using The Malvern Zetasizer.

**Inductively coupled plasma optical emission spectroscopy (ICPOES).** The presence of Eu complexes and PS was determined using inductively coupled plasma optical emission spectroscopy (ICPE-9000, Shimadzu). Europium nitrate and sodium hydrophosphate were used as references.

**Photophysical experiments. The** photophysical characteristics of the **L1**–**L3** ligands and **Eu1**–**Eu3** complexes were measured in distilled dichloromethane. Absorption UV-vis spectra were measured using a Shimadzu UV-1800 spectrometer (Shimadzu, Kyoto, Japan). Excitation and emission spectral data were recorded using a Fluorolog-3 (JY Horiba Inc., Kyoto, Japan) spectrofluorimeter. Quantum yields were calculated using a comparative method using LED 365 nm pumping and [Ru(bpy)_3_](PF_6_)_2_ in water (*Φ_r_* = 0.042) as a standard [61]. The reference refraction indexes were: 1.33 (water), 1.42 (dichloromethane). The equation to calculate quantum yields [62]:(1)$$\Phi_{S}=\Phi_{r}\frac{\eta_{S}^{2}\cdot A_{r}\cdot I_{S}}{\eta_{r}^{2}\cdot A_{S}\cdot I_{r}}$$
where *Φ_S_*—the quantum yield of the sample, *Φ_r_*—the quantum yield of the reference, *η*—the refractive index of the solvent, *A_s_*, *A_r_*—the absorbance of the sample and the reference at the wavelength of excitation of emission, respectively, and *I_s_*, *I_r_*—the integrated area of emission band of the sample and the reference, respectively.

The value of the two-photon absorption cross-section was calculated by finding the value of the two-photon luminescence cross-section [63]:(2)$$\sigma_{\mathrm{TPE}}^{S}=\sigma_{\mathrm{TPE}}^{r}\;\cdot {\left(\frac{n^{S}}{n^{r}}\right)}^{3}\cdot \frac{W^{S}/C^{S}/t^{S}}{W^{r}/C^{r}/t^{r}}$$
where *σ*_TPE_ is the two-photon luminescence cross-section, *σ*_TPA_ is the two-photon absorption cross-section (cm^4^·s), $\sigma_{\mathrm{TPE}}=\Phi \cdot \sigma_{\mathrm{TPA}}$, where *Φ* is the luminescence quantum yield, *W* is the is the number of emitted photons, *C* is the molar concentration of the solution, *n* is the refractive index of the solvent, and *t* is the exposure time in seconds. As the cross-section for two-photon absorption and luminescence, the Goeppert–Mayer value, 10^−50^ cm^4^·s = 1 GM, was used. Fluorescein in aqueous solution at pH = 11 was used as the standard, its quantum yield was considered to be 91% [64], *σ*_TPE_ = 43 GM [63]. The concentration of the studied solutions was ca. 10^−4^ M. The concentration of latex particles in the dispersions under study was ca. 10^−4^% in the units of dry residue mass fraction.

**Quantum yield and lifetime measurements for latex nanoparticles.** The quantum yields of latex nanoparticles in a water dispersion were measured using a Quanta-ϕ integrating sphere with a modular spectrofluorimeter Fluorolog-3 (Horiba Jobin Yvon, Japan). A pulse laser TECH-263 Basic (wavelength, 263 nm; pulse width, 5 ns; and repetition frequency, 10−1000 Hz), an Ocean Optics monochromator (Monoscan-2000; interval of wavelengths, 1 nm), a FASTComTec (MCS6A1T4) multiple-event time digitizer, and a Hamamatsu (H10682-01) photon-counting head were used for lifetime measurements. Cyclic experiments for water dispersion of latex nanoparticles (~0.02 weight% of the dried residue) were performed in the temperature range of 20−40 °C using a temperature-controlled sample compartment qpod 2e (Quantum Northwest, Liberty Lake, WA, USA).

The lifetime vs. temperature calibration curves were obtained using the following protocol: (1) a sample of the probe aqueous dispersion was thermostated at 20 °C for 10 min (the optimized time that it made possible to stabilize the lifetime readings); (2) for the lifetime measurements, the resulting magnitude was the average of 5 times repetition; (3) the temperature was then increased by 5 °C, and the sample was thermostated for 10 min to make 5 independent measurements as above; (4) measurements were repeated in 5-degree increments until the temperature reached 40 °C. 

The protocol for cyclic studies was similar to that used for building up the calibrations. The sensors’ characteristics were calculated in accordance with the literature guidelines [5]. The calculations of temperature sensitivity (*S_T_*), relative sensitivity (*S_r_*), and reproducibility (*R*) are presented in Appendix A.

**Experiments with CHO-K1 cell line.** The Chinese hamster ovary CHO-K1 cells were cultured in DMEM/F12 (Biolot, St. Petersburg, Russia) medium supplemented with 10% FBS (Gibco, Carlsbad, CA, USA), 2 mM glutamine (Gibco, Carlsbad, CA, USA), and penicillin/streptomycin at a concentration of 100 U/mL (Thermo Fisher Scientific, Waltham, MA, USA). The cells were maintained in a humidified incubator at 37 °C with 5% CO_2_ and passaged using trypsin-EDTA (Thermo Fisher Scientific, Waltham, MA, USA). For living-cell confocal microscopy, the cells (1 × 10^5^ CHO-K1 cells in 1 mL DMEM) were seeded in glass-bottomed 35 mm dishes (Ibidi GmbH, Gräfelfing, Germany) and incubated for 24 h. Latex nanoparticles **NPs_Eu2** dispersed in water were dissolved in the growing media and added to the cells at a final concentration of 0.0195 wt.%. The dynamic of internalization was studied at different time points starting from 10 min. Before PLIM cell imaging, the cells were incubated with **NPs_Eu2** for 24 h in the glass-bottom dishes and then the medium was replaced with a fresh one.

**MTT assay.** The Chinese hamster ovary CHO-K1 cells (1 × 10^4^ cells in 100 µL of culture medium/well) were seeded in 96-well plates (TPP, TPP Techno Plastic Products AG, Trasadingen, Switzerland) and incubated overnight. The particles were added to the cells at concentrations of 0.0012–0.039 wt.% for 24 h. The solid content was calculated gravimetrically per 100 mL of water. Then, the cells were treated with the MTT reagent 3(4,5-dimethyl-2-thiasolyl)-2,5-diphenyl-2H-tetrasole bromide (Thermo Fisher Scientific, Waltham, MA, USA) at a concentration of 0.5 mg/mL. Following further incubation at 37 °C under 5% of CO_2_ for 4 h, the formazan crystals were dissolved in DMSO, and the absorbance at 570 nm was measured using a SPECTROstar Nano microplate reader (BMG LABTECH, Ortenberg, Germany). The percentage of viable cells relative to the control was determined for each well as a ratio of the average absorbance value of the wells containing probes to that of the control wells. For each concentration of the particles, the experiment was repeated 12 times. 

**Vital staining of organelles**. LysoTracker Deep Red (Thermo Fisher Scientific, Waltham, MA, USA) was used for the vital staining of acidified compartments, lysosomes, and late endosomes. BioTracker 405 Blue Mitochondria Dye (Merck KGaA, Darmstadt, Germany) was used for the vital staining of mitochondria. The CHO-K1 cells were incubated with **NPs_Eu2** and **NPs_Eu2_TPP** (0.0195 wt.%, 24 h). Then, the cells were rinsed with fresh media 3 × 1 mL, and incubated with a new portion of growing media for 15 min. LysoTracker Deep Red was added to the cells at a concentration of 50 nM and incubated 30 min prior to confocal imaging. BioTracker 405 was added to the cells at a concentration of 50 nM and incubated 15 min prior to confocal imaging.

**Confocal microscopy and PLIM.** Living CHO-K1 cells were imaged by using a confocal-inverted Nikon Eclipse T*i*2 microscope (Nikon Corporation, Tokyo, Japan) with 60× oil immersion objective. The emission of europium was recorded in the range of 663–738 nm using single-photon excitation at 405 nm. In addition to luminescent microphotographs, differential interference contrast (DIC) images were also obtained. The images were processed and analyzed using ImageJ software (National Institutes of Health, Bethesda, MY, USA). The emission of the particles was excited with a 405 nm laser and recorded in the 570–620 nm (red channel) range. The fluorescence of LysoTracker Deep Red was excited at 638 nm and recorded at 663–738 nm. The fluorescence of BioTracker 405 Blue Mitochondria Dye was excited at 405 nm and recorded at 425–475 nm.

Phosphorescence lifetime imaging microscopy (PLIM) of CHO-K1 cells was carried out using the time-correlated single-photon counting (TCSPC). A DCS-120 module (Becker&Hickl GmbH, Berlin, Germany) integrated with the Nikon Eclipse T*i*2 confocal device. A picosecond laser (405 nm) was used as an excitation source. The phosphorescence of the probe was recorded using a 575 longpass filter and 630/75 nm bandpass filter, and a pinhole of 0.5. PLIM images were obtained using the following settings: frame time of 42.87 s, pixel dwell time of 10.46 ms, points number of 1024, time per point of 10.00 µs, time range of PLIM recording of 10.24 ms, total acquisition time of 87–131 s, and image size of 64 × 64 pixels. Oil immersion 60× objective with zoom 7.11 provided a scan area of ca. 0.04 mm × 0.04 mm. Phosphorescence lifetime images were processed using SPCImage 8.1 software (Becker & Hickl GmbH, Berlin, Germany). All PLIM measurements were performed in a humidified Stage Top Incubator Tokai HIT (Fujinomiya, Japan) at 25, 30, or 40 °C and 5% CO_2_. The phosphorescence decay curves were fitted in monoexponential mode. The average number of photons per curve was not less than 10,000 at binning 5–6.

## 4. Conclusions

The obtained novel europium complexes display desirable photophysical properties, namely, excitation with the bio-friendly visible (405 nm) and NIR (800 nm, TPE absorption) radiation and temperature dependence of emission characteristics (intensity and lifetime) in a physiologically relevant interval (20–40 °C). To avoid europium emission quenching in aqueous media, the complexes were successfully embedded into biocompatible latex nanoparticles (**NPs_Eu1**, **NPs_Eu2**, and **NPs_Eu3**) using nanoemulsion polymerization. Additionally, the surface of the NPs was modified with amino functions that provide a potential opportunity for their vectorization aimed at probe-targeted delivery. The photophysical properties of the europium chromophores remain nearly unchanged in the NPs, including lifetime sensitivity to temperature, with the highest one of 1.3%/K found for **NPs_Eu2**. For this probe, the investigation of cytotoxicity, internalization dynamics, and localization in CHO-K1 cells showed that it is possible to apply it safely in cell experiments. We also examined the **NPs_Eu2** lifetime dependence on temperature under different conditions: in aqueous solution, in phosphate buffer, and in a mixture of the growth media (DMEM) and fetal bovine serum (FBS). The obtained data allow building up the calibration curve for temperature estimation in biological samples. PLIM experiments on the CHO-K1 cell line showed that this probe and obtained calibration curve can be used for the estimation of temperature in cells as well as for detection of the cell response to chemical shock, expressed as an increase in intracellular temperature. Unfortunately, vectorization of the probe surface with a phosphonium cation did not give its preferential localization in mitochondria that implies further studies to search for effective ways to deliver the probe to target cell compartments.

## Data Availability

All data reported herein are accompanying the present article.

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
