# Peer review of "Polymeric Nanoparticles with Embedded Eu(III) Complexes as Molecular Probes for Temperature Sensing"

_molecules, 2022, doi:10.3390/molecules27248813_

Round 1

Reviewer 1 Report

  This manuscript introduced a type of polymeric nanoparticles bearing luminescent Eu (III) complexes which were sensitive with temperature. The materials were fully characterized and the NPs-Eu 2 was confirmed to be a successful probe. This work might be suitable for publication in molecules, but there are some comments to improve this work.

1.      It would be better to carry out the cell experiment of NPs-Eu1, with a longer lifetime compared with NPs-Eu2. Because the long lifetime will help to eliminate interference from the potential background noise in the further investigation.

2.      In the experimental details, the assignment of the 1H NMR spectra of the complexes and ligands are missing.

Reviewer 2 Report

In this manuscript, authors synthesized different kinds of polymeric nanoparticles with embedded Eu(III) complexes and used these probes for temperature sensing. This work is interesting. There are some questions should be considered.

(1)    The elemental mapping image of the polymeric nanoparticles should be provided. This could provide the evidence of the existence in Eu.

(2)    In Table 1, the authors summarized the photophysical property of Eu(III) complexes and Eu nanoparticles. In order to provide the advantage of this work, authors should compare them with the other commercially available luminescent probes.

(3)    Please give more discussion about why NPs_Eu2 was selected as the optimal luminescent probes, in comparison with the other Eu nanoparticles.

(4)    In the Figure 4(a), the MTT assay should be analyzed by changing the concentration of NPs_Eu2 in aqueous solution. Authors used the concentration of dried reside to characterize the relationship. Please give some explanation.

(5)    In the Figure 4(b), the MTT assay should be analyzed by using cell viability. Authors used integrated intensity (a.u.) to characterize the relationship. Please give some explanation.

Round 2

Reviewer 2 Report

Accept in present form.